# Long-Term Outcome and Rejection After Allogeneic Uterus Transplantation in Cynomolgus Macaques

**DOI:** 10.3390/jcm8101572

**Published:** 2019-10-01

**Authors:** Iori Kisu, Hirohito Ishigaki, Katsura Emoto, Yojiro Kato, Yohei Yamada, Kentaro Matsubara, Hideaki Obara, Yohei Masugi, Yusuke Matoba, Masataka Adachi, Kouji Banno, Yoko Saiki, Iori Itagaki, Ikuo Kawamoto, Chizuru Iwatani, Takahiro Nakagawa, Hideaki Tsuchiya, Takako Sasamura, Hiroyuki Urano, Masatsugu Ema, Kazumasa Ogasawara, Daisuke Aoki, Kenshi Nakagawa, Takashi Shiina

**Affiliations:** 1Department of Obstetrics and Gynecology, Keio University School of Medicine, Tokyo 1608582, Japan; 2Department of Pathology, Shiga University of Medical Science, Shiga 5202192, Japan; 3Department of Pathology, Keio University School of Medicine, Tokyo 1608582, Japan; 4Department of Surgery, Kidney Center, Tokyo Women’s Medical University, Tokyo 1628666, Japan; 5Department of Pediatric Surgery, Keio University School of Medicine, Tokyo 1608582, Japan; 6Department of Surgery, Keio University School of Medicine, Tokyo 1608582, Japan; 7Department of Anesthesiology, Saiseikai Kanagawaken Hospital, Kanagawa 2210821, Japan; 8Research Center for Animal Life Science, Shiga University of Medical Science, Shiga 5202192, Japan; 9The Corporation for Production and Research of Laboratory Primates, Ibaraki 3050003, Japan; 10Safety Research Center, Ina Research Inc., Nagano 3994501, Japan; 11Department of Molecular Life Science, Division of Basic Medical Science and Molecular Medicine, Tokai University School of Medicine, Kanagawa 2591193, Japan

**Keywords:** uterus transplantation, uterine transplantation, cynomolgus macaque, uterine factor infertility, allogeneic uterus transplantation, rejection

## Abstract

Uterus transplantation (UTx) is an option for women with uterine factor infertility to have a child, but is still in the experimental stage. Therefore, allogeneic animal models of UTx are required for resolution of clinical issues. In this study, long-term outcomes were evaluated in four recipients (cases 1–4) after allogeneic UTx in cynomolgus macaques. Immunosuppression with antithymocyte globulin induction and a triple maintenance regimen was used. Postoperative ultrasonography and biopsy of the transplanted uterus and immunoserological examinations were performed. All four recipients survived for >3 months after surgery, but continuous menstruation did not resume, although temporary menstruation occurred (cases 1 and 2). All animals were euthanized due to irreversible rejection and no uterine blood flow (cases 1, 2 and 4) and post-transplant lymphoproliferative disorder (case 3). Donor-specific antibodies against MHC class I and II were detected in cases 1, 2 and 4, but not in case 3. Peripheral lymphocyte counts tended to elevate for CD3^+^, CD20^+^ and NK cells in conjunction with uterine rejection, and all animals had elevated stimulation indexes of mixed lymphocyte reaction after surgery. Establishment of allogeneic UTx in cynomolgus macaque requires further exploration of immunosuppression, but the clinicopathological features of uterine rejection are useful for development of human UTx.

## 1. Introduction

Uterus transplantation (UTx) has become a potential option for women with uterine factor infertility to have a child. Data accumulated from animal studies for more than a decade have led to clinical application of UTx. Brännström et al. described the first human delivery after UTx with a living donor in 2014 [1] and Ejzenberg et al. achieved the first live birth after UTx from a deceased donor in 2017 [2]. These important achievements have attracted considerable attention worldwide, and many teams have recently aimed for clinical application of UTx with accumulation of basic experimental data. However, UTx is still in an experimental stage and there are many clinical and technical issues to be resolved [3]. Therefore, careful accumulation of data in animal experiments is needed for full establishment of clinical application of UTx because the technology associated with UTx has not been standardized.

Animal studies on UTx have been performed in mice, rats, rabbits, dogs, pigs, sheep, and non-human primates [4]. Delivery after allogeneic UTx has only been achieved in rats [5] and sheep [6]. There are only a few reports on UTx in non-human primates, and delivery after allogeneic UTx in these species has yet to be attained because of specific problems and limitations [7,8]. However, UTx studies in non-human primates are likely to provide important preclinical data for resolution of problems in UTx in humans because these animals are anatomically and physiologically similar to humans.

Our team in Japan has accumulated a large UTx research data archive using cynomolgus macaques [9], which are commonly used as experimental models because of their anatomic and physiologic similarity to humans. This allows data to be extrapolated to humans, except that both the superficial and deep uterine veins connect to the external iliac veins in cynomolgus macaques [10]. Also, the body size of female cynomolgus macaques used in our laboratory is similar to that of human neonates and the weight is approximately 2.5–4 kg, which is a significant disadvantage compared with other non-human primate models, such as baboons [11]. Therefore, the challenge with this species is the need for delicate surgical techniques for dissection and vascular anastomosis for their smaller vessels. We previously succeeded in delivery after autologous UTx in cynomolgus macaque and established an autologous UTx animal model [12]. Thereafter, we tried to establish an allogeneic UTx model with living donors [13], but we encountered many cases of massive intraoperative bleeding and death due to hemorrhage intraoperatively and postoperatively.

With this background, we decided to move on to establishment of a surgical technique for allogeneic UTx, assuming recovery of a uterus from a deceased donor, to enable procurement of wider and longer vessels; consequently, the surgical procedures and model preparation are more stable, in comparison with a living donor. As described in a previous report [14], we succeeded in short-term survival after allogeneic UTx and establishment of a stable allogeneic UTx model with procurement of common iliac vessels or aorta and inferior vena cava from a donor cynomolgus macaque. However, this report had the important limitation that long-term outcomes, including immune system suppression and postoperative organ rejection, were not evaluated. In preliminary allogeneic UTx with living cynomolgus macaque donors [13], we were unable to overcome uterine rejection after surgery. Therefore, as a next step for establishment of an allogeneic UTx model, long-term follow-up and use of immunosuppressants are important. Herein, we report long-term outcomes in the four cynomolgus macaques with short-term survival after allogeneic UTx described in our previous report [14].

## 2. Materials and Methods

### 2.1. Animals

Four female cynomolgus macaques (*Macaca fascicularis*, age 5–9 years; average body weight, 3.83 ± 1.12 kg (mean ± standard deviation) after allogeneic UTx [14] were used in this study. The four recipients and their donors were selected for compatible ABO blood type and a high degree of polymorphism in the major histocompatibility complex (MHC) gene. Negative blood type cross match and donor-specific antibody (DSA) before surgery between donor and recipient were confirmed. The study was performed in accordance with the recommendations in the Guide for the Care and Use of Laboratory Animals of the National Research Council and approved by the Animal Care and Use Committee of the Research Center for Animal Life Science, Shiga University of Medical Science, Japan (permit number: 2013-4-2).

### 2.2. MHC Typing

Total RNA was isolated from peripheral blood cells using TRIzol reagent (Invitrogen/Life Technologies/Thermo Fisher Scientific, Carlsbad, CA, USA). Complementary DNA was synthesized using ReverTra Ace (Toyobo, Osaka, Japan) after treatment of isolated RNA with DNase I (Invitrogen/Life Technologies/Thermo Fisher Scientific). Six primer pairs for MHC class I (Mafa-A and Mafa-B), Mafa-DRB, Mafa-DQA1, Mafa-DQB1, Mafa-DPA1 and Mafa-DPB1 loci were used separately for RT-PCR amplification [15,16]. In addition to these primer sets, we also designed 50 fusion primers that contained the 454 Titanium adaptor, key and multiple identifier sequences in each locus. After RT-PCR amplification using high fidelity KOD FX polymerase (Toyobo), massively parallel pyrosequencing of the PCR products was carried out using the GS Junior system and amplicon sequencing protocol (Roche, Basel, Switzerland). MHC genotypes were assigned by comparing sequences to known MHC allele sequences in the Immuno Polymorphism Database (http://www.ebi.ac.uk/ipd/index.html) [17] using GS Reference Mapper ver. 3.0. The mapping parameter was set to a perfect match between the read and reference sequences to avoid mismapping among Mafa loci and contamination of in vitro generated PCR crossover amplicons [18]. If short-read sequences were not mapped to a reference sequence, a consensus sequence output from GS Reference Mapper ver. 3.0 was used as a reference sequence for secondary mapping [19]. MHC genes of the four recipients and donor pairs are shown in Appendix A.

### 2.3. Uterus Transplantation

Operative anesthesia and surgical procedures in donors and recipients were performed as previously described [14]. In brief, animals were intubated and operated on under general anesthesia with ventilation maintained by isoflurane inhalation (0.5%–1.5%; Abbot Japan, Tokyo, Japan). All animals received 25 mg/kg of cefazolin every 3 h as antibiotic prophylaxis from initiation of the operation. In donor surgery, thoracolaparotomy was performed, and the iliac vessels and those surrounding the uterus were dissected in cases 1 and 2. For further improvement of the surgical procedure based on the experience in cases 1 and 2, additional dissection including the abdominal aorta/inferior vena cava (IVC) above the left renal vein and ovarian veins was performed in cases 3 and 4. The ovaries of donors were removed in situ, but oviducts of donors were not removed. An anticoagulant (heparin, 1000 IU) was administered intravenously 5 min before cross-clamping of the abdominal aorta. The uterine grafts were perfused with 200–350 mL of cold histidine-tryptophan-ketoglutarate solution (HTK^®^; Custodiol^®^, Nordmedica, Gentofte, Denmark) via a perfusion catheter (22-G intravenous needle; Terumo Corp., Tokyo, Japan) in the femoral or external iliac artery. Then, en bloc retrieval of the uterus with the adnexa, ovarian vein, and proximal vagina was performed, including the iliac vessels (cases 1 and 2) or infrarenal aorta/IVC (cases 3 and 4). The ovarian veins draining to the renal vein or IVC were left in the uterine graft side in cases 3 and 4. In recipient surgery, after entering the peritoneal cavity through a midline incision, a total simple hysterectomy was performed in advance, with preservation of the ovaries. Subsequently, iliac vessels on both sides (cases 1 and 2) or the infrarenal abdominal aorta/IVC (cases 3 and 4) were exposed at a distance of approximately 2 cm. The uterine graft was brought into the operative field and the vagina of the graft was anastomosed with the vaginal vault of the recipient to fix the uterus in the pelvis. Before vascular anastomoses, heparin (100 IU) was administered intravenously. In recipient surgery, end-to-side anastomosis with the recipient’s and donor’s common iliac vessels (cases 1 and 2) or infrarenal aorta/ICV (cases 3 and 4) was performed using continuous sutures (7–0 or 8–0 Prolene^®^, Johnson & Johnson; 9–0 Asflex^®^, Kono Seisakusho, Co., Ltd., Tokyo, Japan). After reperfusion, sutures of the retroperitoneum and round ligaments were performed before closing the abdominal incision. Normothermia was maintained throughout the operation using heating lights and blankets in donors and recipients. Antibiotics and a proton-pump inhibitor were administered after surgery.

### 2.4. Intraoperative Indocyanine Green Fluorescent Angiography

Intraoperative indocyanine green (ICG) (Diagnogreen 0.5%; Daiichi Pharmaceutical, Tokyo, Japan) was injected intravenously to confirm the blood supply in the transplanted uterus and blood flow at the site of vascular anastomosis during surgery and in second-look surgery or autopsy [20,21,22]. Fluorescence intensity was displayed using a Photodynamic Eye Neo system (Hamamatsu Photonics K.K., Hamamatsu, Japan).

### 2.5. Immunosuppressive Management

Severe rejections occurred in the first two transplants (cases 1 and 2) and the immunosuppressive protocol was amended in cases 3 and 4. As induction treatment, animals received antithymocyte globulin (ATG) (10 mg/kg; Thymoglobulin, Genzyme, Cambridge, MA, USA) intravenously on postoperative day (POD) 0 (the day of surgery) in cases 1 and 2, and 20 mg/kg of ATG on POD 0 and POD 2 in cases 3 and 4. Maintenance treatment consisted of cyclosporine (CyA) (Sandimmune: Novartis, Basel, Switzerland) given subcutaneously in cases 1 and 2, and tacrolimus (TAC) (Prograf; Astellas Pharma, Tokyo, Japan) given intramuscularly in cases 3 and 4. These drugs were administered from day 0 in case 1, day –2 in case 2, and day –1 in cases 3 and 4. The target trough levels for CyA and TAC up to 1 month after surgery were planned to be in the ranges of 300–400 ng/mL and 15–20 ng/mL, respectively, and thereafter were adjusted as required. As another maintenance treatment, mycophenolate mofetil (MMF) (40–100 mg/kg; Cellcept; Chugai Pharmaceutical, Tokyo, Japan) was administered orally from recovery of appetite after surgery on POD 11 in case 1, POD 6 in case 3, and POD 6 in case 4. MMF was discontinued on POD 22 in case 1 because the animal did not tolerate the drug. For this reason, MMF was not given in case 2, but MMF was administered forcibly using an orogastric catheter in cases 3 and 4, based on the severe rejections in cases 1 and 2. Methylprednisolone (10 mg/kg; Solu-Medrol; Pfizer, NY, USA) was injected intravenously on POD 0 and then injected intramuscularly daily starting on POD 1. The dose of methylprednisolone was gradually tapered, and administration was completed on POD 14 in cases 1 and 2. In cases 3 and 4, methylprednisolone was administered at daily doses of 0.5 and 0.2 mg/kg, respectively, until the end of observation. If rejection occurred, steroid pulse therapy of 10 mg/kg methylprednisolone was administered and gradually tapered.

### 2.6. Flow Cytometry

After lysing red blood cells with BD Lysing buffer (BD Bioscience, Cat#:555899, Franklin Lakes, NJ, USA), single cell suspensions of peripheral blood cells were immunostained with fluorescence-conjugated antibodies (Appendix A). Dead cells were excluded using 7–aminoactinomycin D (7–AAD; BD Pharmingen, Cat#: 559925, Franklin Lakes, NJ, USA). Isotype-matched antibodies corresponding to each specific antibody were used as negative controls. Samples were analyzed using a CytoFlex S instrument (Beckman Coulter Inc., Cat#: B75408, Brea, CA, USA).

### 2.7. Mixed Lymphocyte Reaction (MLR)

Heparinized peripheral blood was collected from macaques before and after transplantation. After lysing red blood cells, 5 × 10^5^ cells per well were cultured in an ELISPOT plate (Mabtech AB, Cat#: FS-2122-2, Nacka Strand, Sweden) with 5 × 10^5^ donor splenocytes. The culture was usually performed in triplicate wells, but duplicate culture was used when the number of cells was insufficient. After culture for 3 days, the number of interleukin-2 (IL-2) positive spots was counted by an ImmunoSpot analyzer (Cellular Technology, Shaker Heights, OH, USA). Stimulation indices (SI) were calculated by the following formula: the sum of the number of spots in culture of blood cells and donor splenocytes, divided by the number of spots in culture of blood cells only.

### 2.8. Donor-Specific Antibodies in Plasma of Macaques after Transplantation

After heat-inactivation, plasma collected from the transplanted macaques was diluted 10 times with phosphate-buffered saline (PBS; Nacalai Tesque Inc., Cat#: 14249–25, Kyoto, Japan) before use. The diluted plasma was added to a cell suspension of donor splenocytes as a primary antibody. Fluorescein isothiocyanate (FITC)-conjugated goat polyclonal anti-mouse IgG (Nordic Immunological Laboratories, Cat#: GAM/Fab/FITC, Copenhagen, Denmark) was used as a secondary antibody to detect the attached IgG on the surface of the donor cells. Phycoerythrin conjugated anti-CD20 antibody (Clone: 2H7, BioLegend, Cat#: 302306, San Diego, CA, USA), and allophycocyanin conjugated anti-CD3 antibody (Clone: SP34-2, BD Pharmingen, Cat#: 557597) were also used to study the cell type to which the donor specific IgG attached. The cells were analyzed by FCM.

### 2.9. Postoperative Assessment

After the surgical procedure, animals were separately accommodated in a single cage and the general condition (appetite, bowel movement, vomiting, urination, attitude) and laparotomy wound site were evaluated daily. Laboratory assessments including hematology and blood chemistry were performed three times per week for the first two postoperative weeks, two times per week for two months, and weekly thereafter. To monitor for potential rejection after surgery, the size of the transplanted uterus and blood flow in the transplanted uterine artery were determined by transabdominal ultrasonography under anesthesia. Transvaginal biopsy of the transplanted uterine cervix and body and FACS, DSA and MLR analyses were routinely conducted monthly or when considered necessary (whenever graft rejection was suspected). Animals were euthanized if severe weakness, weight loss or abnormal behavior was seen or when determined to be necessary by veterinary staff and investigators. Necropsy was performed immediately after euthanasia to evaluate the graft and extrapelvic organs, including the brain, lung, heart, thymus, thymus, liver, gallbladder, spleen, pancreas, stomach, duodenum, jejunum, ileum, colon, rectum, kidney, bladder, bone marrow and lymph node.

### 2.10. Histological Evaluation, Immunohistochemistry and in Situ Hybridization

Formalin-fixed, paraffin-embedded liver biopsy tissues were sliced to a thickness of 3 μm and stained with hematoxylin and eosin (H&E). Elastica van Gieson (EVG) staining was also performed. An automated staining system (Bond Max, Leica Biosystems) with a Bond Polymer Refine Detection kit (Leica Biosystems; catalog #: DS9800) was used for immunohistochemistry and in situ hybridization. Immunohistochemistry was performed using an anti-CD20 monoclonal antibody (mouse; clone L26; dilution, 1:200; Leica Biosystems; catalog #: CD20-L26-L-CE-H), an anti-CD3 monoclonal antibody (rabbit; clone SP7; dilution, 1:4; Abcam; catalog #: ab21703), and an anti-CD8 polyclonal antibody (rabbit; dilution, 1:400; Abcam; catalog #: ab4055). An EBER Probe (ready to use; Leica Biosystems; catalog #: PB0589) was used for detection of Epstein–Barr virus-encoded mRNA by in situ hybridization. Allograft rejection was evaluated histopathologically using a grading system for monitoring of rejection proposed by a Swedish group [23,24], in which rejection of grades 1, 2 and 3 is described as mild, moderate and severe, respectively.

## 3. Results

### 3.1. Changes of Trough Levels of Immunosuppressants

Changes of trough levels of CyA (cases 1 and 2) and TAC (cases 3 and 4) in maintenance treatment are shown in Figure 1.

### 3.2. Overall Outcome

All four recipient animals survived for more than 3 months after surgery without any surgical complications. The overall outcomes in these animals are summarized in Table 1.

### 3.3. Case 1

Oral administration of MMF (50 mg/kg, twice a day) was started on POD 11 because of recovered appetite, but discontinued on POD 22 due to a decrease in appetite. On POD 32, an increased inflammatory response (WBC 181 × 10^2^/µL, CRP 11.8 mg/dL) and elevated LDH (1076 IU/L) were found. Swelling of the uterus was found despite blood flow in the uterine artery identified by transabdominal ultrasonography, and rejection was suspected. However, rejection could not be confirmed because samples that were appropriate for evaluation were not obtained from cervical biopsy specimens. The increased inflammatory response and LDH were improved by antibiotics; however, a biopsy specimen from the uterine cervix on POD 70 showed stromal lymphocyte infiltration with capillary endotheliitis, which was consistent with severe rejection (Figure 2A). Consequently, steroid pulse therapy was used.

A second-look surgery was performed on POD 133 to assess the therapeutic effect. Subsequent ICG fluorescence angiography showed no uterine blood flow, and a uterine biopsy specimen showed coagulative necrosis, fibrosis with hyalinization, and lymphocyte infiltration with endotheliitis in the myometrial tissue. Immunohistochemistry showed that CD8^+^ T-cells had infiltrated into capillary vessels. (Figure 2B). We interpreted these findings as severe rejection with resistance to immunosuppressive treatment. Administration of CyA was discontinued on POD 143. Systemic conditions were good, but the uterus gradually shrank and was dissected on POD 196. Intraperitoneal findings showed a white shrunken uterus adhered to a wide range of the omentum and rectum, with left hydrosalpinx (Figure 2C). In vascular anastomotic sites, the grafted common iliac artery had collapsed and no beat was detected. ICG fluorescent angiography showed no blood flow in the whole uterus and in anastomosed grafted vessels (Figure 2D). Histologically, the uterus was severely atrophic and had no endometrium. The myometrial area was replaced by hyalinized fibrosis in which small vessels were occlusive. The main grafted vessels showed complete fibrous occlusion, CD8-rich lymphocyte infiltration, and hemosiderin deposition (Figure 2E). The donor’s grafted oviduct showed dilation and CD8-rich lymphocyte infiltration, which also suggested rejection (Figure 2F).

### 3.4. Case 2

Administration of MMF was not planned in case 2 because of the postoperative decrease in appetite in case 1. Blood flow in the uterine artery was identified by transabdominal ultrasonography on POD 20, but slight swelling of the uterus was found. However, menstruation resumed and follow-up was performed without treatment. Biochemistry on POD 32 showed an increased inflammatory response (WBC 195 × 102 uL, CRP 14.6 mg/dL) and elevated LDH (918 IU/L). Rejection was suspected based on the results of case 1. Consequently, second-look surgery was performed on POD 35. The uterus was grossly adhered to the omentum and was markedly swollen (Figure 3A). High adhesion was found around the adnexa. Histologically, myometrial tissue showed focal necrotic change. Endotheliitis with CD8-rich lymphocyte infiltration and endothelial desquamation were observed in the myometrial interstitium (Figure 3B). These findings suggested severe rejection. Consequently, steroid pulse therapy and administration of ATG (20 mg/kg) were performed.

Pathological rejection including myometrial necrosis and endotheliopathy was still found in a uterine biopsy on POD 70, indicating no therapeutic effect. Based on the above, improvement of rejection was determined to be difficult, and administration of CyA was discontinued on POD 75. No uterine blood flow was found on transabdominal ultrasonography on POD 97 and abscesses were present in the uterine cavity (Figure 3C). Transvaginal drainage was performed and *Escherichia coli* was identified in a bacterial culture test. A slightly elevated inflammatory response (WBC 111 × 10^2^ uL, CRP 1.7 mg/dL) was found in blood tests, but systemic conditions were good. After that, the uterus gradually shrank and was dissected on POD 126. Similarly to case 1, the intraperitoneal findings were a white shrinking uterus adhered to a wide range of the greater omentum and rectum, with bilateral hydrosalpinx (Figure 3D). Vascular anastomotic sites could not be identified due to high adhesion. The whole uterus was not imaged by ICG fluorescence angiography (Figure 3E). The removed uterus showed atrophy with hyaline fibrosis. In the fibrotic area, CD8-rich lymphocyte infiltration was observed. The uterine cavity contained a lot of neutrophils, which was suggestive of uterine infection (Figure 3F). As for case 1, the vessels around grafted tubes showed CD8-rich endotheliitis.

### 3.5. Case 3

MMF was administered twice a day at 50 mg/kg using an orogastric catheter from POD 6. Blood flow in the uterine artery was normal in ultrasound on POD 20 and no rejection was found in biopsy. However, severe anemia occurred with Hb down to 4.4 g/dL from POD 38–61, as an adverse effect of MMF. Consequently, MMF was withdrawn and transfusion was performed to treat anemia. During that period, trough concentrations of tacrolimus were low (Figure 1). Biochemistry on POD 61 showed an increased inflammatory response (WBC 169 × 10^2^ uL, CRP 20.8 mg/dL) and elevated LDH (515 IU/L), and rejection was suspected based on the results in cases 1 and 2. Consequently, the dose of tacrolimus was increased from POD 62, MMF was readministered at 30 mg/kg twice a day, and steroid pulse therapy was started. The inflammatory response and LDH improved, but the trough concentrations of tacrolimus were poorly controlled and high until POD 116 (Figure 1). Uterine blood flow was poor in ultrasonography on POD 83.

A uterine biopsy on POD 83 showed stromal CD8-rich lymphocyte infiltration and liquefaction degeneration with a small number of Civatte bodies (Figure 4A,B). The biopsy also included myometrial tissue, which was atrophic and had a small area of hyalinization. These findings indicated at least moderate rejection, and this was improved to mild rejection based on a biopsy on POD 118. However, the macaque developed right eyelid swelling on POD 120 (Figure 4C) and bilateral leg paralysis on POD 129, resulting in buttock decubitus due to leg paralysis. Consequently, the macaque was euthanized and dissected on POD 140. Laparotomy and thoracotomy results showed a red-colored uterus of normal size and adhesion around the left adnexa (Figure 4D). Tumors of 4 cm and 7 cm were found near the abdominal aorta and the anterior mediastinum, respectively, and tumors were also found in the left adrenal gland and right eyelid. Histologically, the uterine cervix showed mild lymphocyte infiltration, liquefaction degeneration, and a small number of Civatte bodies, which still indicated mild rejection. However, all nodular tumors were composed of large-sized atypical B-cells and focal necrotic changes. These tumor cells were positive for CD20 and EBER-ISH, but negative for CD3 and CD8. The diagnosis was monomorphic post-transplant lymphoproliferative disorder (PTLD), diffuse large B-cell lymphoma (Figure 4E,F). Possible causes of bilateral leg paralysis were spinal cord ischemia and stenosis induced by vascular retraction by PTLD and direct invasion of the spinal cord. Serum EB virus IgG antibody was positive before UTx.

### 3.6. Case 4

Based on the PTLD in case 3, the dose of MMF on POD 6 was decreased from 40 to 20 mg/kg (twice a day, using an orogastric catheter) and the maintenance dose of methylprednisolone was decreased from 0.4 to 0.2 mg/kg in case 4. When starting administration of ATG, ganciclovir (20 mg/kg; Denosine; Mitsubishi Tanabe Pharma, Osaka, Japan) was also injected intravenously. Transabdominal ultrasonography on POD 13 showed a uterus of normal size and good blood flow in the uterine artery. Histologically, liquefaction degeneration was mild; however, perivascular lymphocyte infiltration and endotheliitis were clearly evident. Based on the endotheliitis, this condition was diagnosed as moderate rejection. No increased inflammatory response or LDH was found in biochemistry tests. Consequently, steroid pulse therapy was used. Transabdominal ultrasonography on POD 25 showed slight uterine swelling. A biopsy specimen showed similar rejection features to those on POD 13, with perivascular fibrosis and edematous changes. Steroid pulse therapy was administered again, ATG was given at 20 mg/kg, and MMF was increased to 30 mg/kg twice a day. In a biopsy on POD 41, obstructive venulitis and endotheliitis of small to middle size arteries were seen, in addition to worsening of capillary endotheliitis, and perivascular fibrosis (Figure 5A). Biochemistry findings on POD 65 showed an elevated inflammatory response (WBC 206 × 10^2^ uL, CRP 17.1 mg/dL). This was improved by antibiotics, leading to a good systemic condition.

The uterus subsequently became more swollen and was dissected on POD 104, after it was decided that rejection could not be relieved. Laparotomy results showed a markedly swollen white uterus that adhered to the omentum and to vascular anastomotic sites surrounding the aorta and IVC (Figure 5B). The whole uterus was not imaged, but blood flow at vascular anastomotic sites distal from the aorta was not detected by ICG fluorescence angiography. A hypertrophic myometrium was found in the resected uterus, and no endometrium was detected. An abscess was found in the uterine cavity and *Escherichia coli* was identified in a bacterial culture test. In the resected aorta, the donor-grafted intravascular space in the vascular anastomotic sites was almost stenosed (Figure 5C) and thrombi were found in the grafted intravascular space from the aorta to bilateral common iliac arteries (Figure 5D). Severe endotheliitis was observed in the whole uterus, along with epithelial desquamation in the cervix (Figure 5E), obstructive vasculitis and focal necrosis in the corpus. The donor’s oviducts had severe lymphocyte infiltration and epithelial apoptotic changes. In the grafted aorta, lymphocytes were not attached to the endothelium of the aorta directly; however, its feeding vessels in the vascular adventitia showed severe endotheliitis, mainly due to CD8^+^ T-cells, and some showed obstruction (Figure 5F).

## 4. Recovery of Cyclicity

Temporary menstruation resumed from POD 45 and POD 96 in case 1 and POD 17 in case 2, but amenorrhea then continued in both animals. Menstruation did not resume after surgery in cases 3 and 4.

### 4.1. Changes in Peripheral Lymphocyte Counts

Induction treatment and additional ATG administration decreased lymphocyte counts immediately after surgery (Figure 6). Due to differences in ATG doses, this decrease in cases 3 and 4 was prolonged compared to that in cases 1 and 2. Changes of peripheral CD3^+^, CD20^+^ and NK cell (CD3^–^ CD8^+^ CD16^+^) counts and the CD4^+^/CD8^+^ cell ratio are shown in Figure 7. Immediately after induction treatment, counts of CD3, CD20 and NK cells decreased, but then gradually increased in correspondence to rejection. The CD4^+^/CD8^+^ ratio showed a tendency to decrease. Regulatory T cells (CD4^+^ FOXP3^+^) had no relationship with rejection (data not shown).

### 4.2. Leukocyte Proliferation Against Donor or Third-Party Antigens

The MLR stimulation index against donor and third-party antigens was determined after surgery (Figure 8). All animals exhibited elevation of this index.

### 4.3. Antidonor Antibody Production

All animals were seronegative for DSA before transplantation. DSAs against MHC class I and II were detected in conjunction with uterine rejection on PODs 46 (data not shown), 35 and 13 in cases 1, 2 and 4, respectively, after which the DSA titer increased (Figure 9). In case 3, DSAs against MHC class I and II were negative.

## 5. Discussion

UTx is now conducted in humans in several countries, but is still considered to be in the experimental stage since there are remaining clinical and technical issues to be resolved [25]. Therefore, development of allogeneic animal models of UTx is required for improvement of clinical UTx, including establishment of suitable immunosuppressants, diagnosis of rejection, monitoring of a grafted uterus, uterine antigenicity, and teratogenicity induced by immunosuppressants. Experiments in non-human primates are particularly appropriate for extrapolation to clinical application of UTx because these animals are anatomically and physiologically similar to humans. However, there are some problems and limitations with experimental use of non-human primates for UTx [7,8]. Female macaques are small compared to humans, and complex vascular dissection of the pelvic floor is required in UTx. Therefore, fine surgical techniques are required in a macaque model. Postoperative management also has difficulties. Control of the blood level of immunosuppressive agents is important because of effects on graft survival, but it is difficult to administer these agents orally in non-human primates as scheduled, and delivery via enteral feeding is problematic because of marked anorexia after invasive surgery and side effects of the agents. Intravenous administration is simple in humans, but difficult to control in animal models. These difficulties also cause marked postoperative weight loss that may not be recovered by supplementation with oral nutrients alone, and too much weight loss leads to an endpoint of the study in most settings. Moreover, postoperative examinations including echography and biopsy cannot be easily performed because sedation is required, and blood test items are also limited in non-human primates, compared to humans.

In this study, long-term outcomes after allograft were examined in four cynomolgus macaques used in our previous study to establish the allograft procedure. All four macaques developed rejection and three underwent hysterectomy because uterine function could not be recovered due to irreversible chronic rejection. There are case reports of human UTx with rejection [23,26,27,28] but no case of hysterectomy due to refractory rejection. Therefore, clinicopathological features of rejection in the macaques in follow-up are important findings for solving problems from the perspective of basic medicine in UTx. The clinical features of severe rejection found in this study are summarized in the following paragraphs.

First, an increased WBC and CRP inflammatory response due to rejection and elevated LDH due to cell collapse were found. In rejection or elimination of conventional life-supporting organ transplants, clinical symptoms and biochemical data change immediately. In contrast, the uterus is not a life-supporting organ and rejection is not immediately life-threatening and frequently asymptomatic; therefore, there are no clear findings showing rejection. The biochemical findings above may be one indicator; however, these findings are often caused by irreversible rejection and it may be difficult to recover uterine function in such a case.

Second, the uterus is swollen by rejection and then shrinks over time. In the course, inflammatory edema occurs due to rejection, followed by cell injury and necrotic cell death, and granulation tissues and fibrils are formed, resulting in atrophy. In cases 1 and 2, these histopathologic findings were found, but in case 4, the uterus swelled but did not shrink. However, endotheliopathy and vascular occlusion were observed, and the uterus was assumed to have atrophied due to ischemia over time. This assumption is suggested by our previous results showing that the uterus of cynomolgus macaque shrinks after a warm ischemia time of 8 h [29,30].

Third, continuous rejection sometimes promotes intrauterine infection. The uterine cavity is in contact with the outside of the body through the vagina; therefore, intrauterine infection may occur. Since organ transplant recipients are treated with immunosuppressant drugs, they have higher risks for infection than general patients. Furthermore, rejection and ischemia disrupt phylaxis functions in the uterus and enhance the risk for infection. In cases 2 and 4, intrauterine infection occurred, and *E.* coli was detected in both. Hygiene control of reared cynomolgus macaques is limited and *E. coli* from feces probably contaminated the uterus. Also, a UTx patient from Sweden had repeated intrauterine infection and the uterus had to be resected [31].

Fourth, rejection causes thrombus formation. An intravascular thrombus is often caused by surgical procedures of vascular anastomosis and sometimes by blood flow stasis because of occlusion of small and middle-size vessels in the rejected organ due to rejection-induced endotheliopathy. In case 4, thrombi were found from the vascular anastomotic site of the grafted aorta to the iliac artery and severe endotheliopathy occurred. Furthermore, inflammatory cells were detected in the outer membranes of the grafted aorta, which suggests that rejection in vessels is probably the cause of thrombi.

Fifth, interestingly, rejection occurred in the oviducts, as well as in the uterus. In cases 1 and 2, lymphocytic infiltration was found in the oviducts of donors. In human UTx, the donor’s oviducts are often removed in transplantation, despite retention of the oviduct in donors having an advantage of maintaining the option of natural pregnancy. However, hydrosalpinx was evoked in cases 1 and 2 due to the remaining oviduct, and this may cause infertility; therefore, surgery with oviduct resection is currently preferable.

Finally, the histopathological findings in rejection due to UTx in cynomolgus macaques included frequent onset of endotheliopathy, in addition to those based on clinical criteria for uterine rejection by a Swedish group [23,24]. Simultaneously, DSAs against MHC class I and II are likely to be positive. Therefore, additional induction treatment with rituximab and a triple regimen may be necessary to inhibit B cells from humoral rejection. Histopathology showed endotheliopathy in the interstitium more than the epithelium; therefore, humoral rejection may be overlooked if interstitial biopsy is not performed.

Postoperative immunosuppression for prevention of rejection due to organ-specific antigenicity is important in organ transplantation. Antigenicity of the uterus is unclear, but in human UTx many patients have received immunosuppressant protocols similar to those used in renal transplantation. There have been no reports of serious rejection of the uterus; therefore, the uterus is considered to be an organ that is unlikely to develop rejection. However, from our previous results [13] and the outcomes of this study, we conclude that the uterus of cynomolgus macaques is likely to be rejected. Furthermore, in transplantation in cynomolgus macaques, higher doses of immunosuppressants than those used in humans are required, and it is necessary to increase the blood trough concentrations of calcineurin inhibitors to higher than those in humans [32,33,34,35] or inhibiting T cells with a higher ATG dose [36]. Also, enhanced immunosuppression may cause PTLD, as in case 3. Therefore, strict postoperative control of immunosuppressants with limited drug administration and examination in cynomolgus macaques is difficult.

Our results show that studies in non-human primate models have specific problems and limitations, and recognition is required that that results may not always correspond directly to those in humans. However, development of an allogeneic UTx model in a non-human primate is still important for addressing clinical issues in human UTx, and further validation and outcomes of UTx in such models will contribute to successful clinical use of UTx, in which current results indicate exciting developments. The outcomes of these procedures are likely to be improved by accumulation of data from further animal studies, including in non-human primates.

In conclusion, allograft of the uterus of cynomolgus macaques resulted in clinicopathological findings of increased inflammatory response, swollen and subsequently shrunken uterus, intrauterine infection, intravascular thrombus formation, oviductal rejection, and a DSA-positive status. Postoperative management and monitoring of macaques are limited, but the uterus of cynomolgus macaques was found to be likely to develop rejection and to have high antigenicity. Therefore, adjustment of protocols for immunosuppressive drugs may be necessary to produce a stable uterine allograft model in cynomolgus macaques. However, the current results are important for resolution of remaining clinical questions in human UTx by producing a scientifically verifiable model of UTx in a non-human primate.

## Figures and Tables

**Figure 1 jcm-08-01572-f001:**
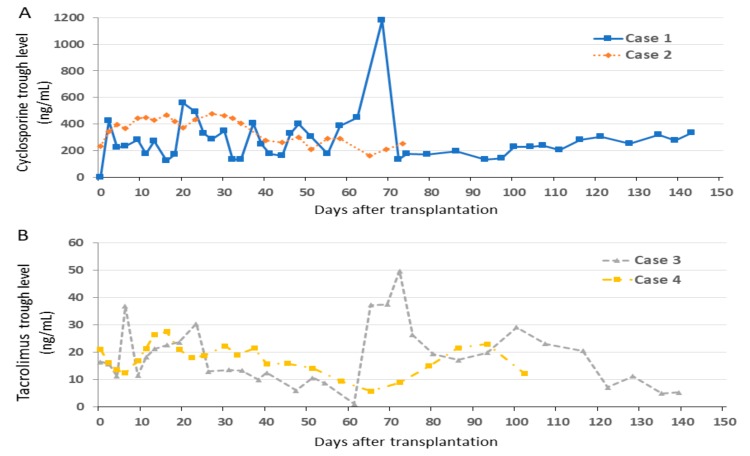
Changes of trough level in immunosuppressive treatment. (**A**) Cyclosporine concentrations in cases 1 and 2. (**B**) Tacrolimus concentrations in cases 3 and 4.

**Figure 2 jcm-08-01572-f002:**
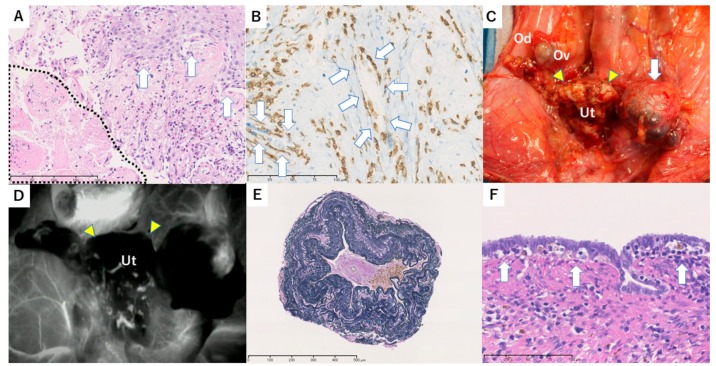
Macroscopic and histopathological findings in case 1. (**A**) Histopathological findings of a biopsy specimen from the uterine cervix on postoperative day (POD) 70. Severe stromal inflammation and capillary endotheliitis including lymphocytes and eosinophils were observed. Lymphocytes also infiltrated into squamous epithelium and resulted in liquefaction degeneration (white arrow). Necrotic change was also seen on the left side (within dotted line). These findings suggest severe rejection. Hematoxylin and eosin (H&E) stain, 200 × bar = 200 µm. (**B**) Histopathological findings of a biopsy specimen from the uterine body in second-look surgery on POD 133. In the necrotic and hyalinizing areas of the uterus, CD8-rich lymphocyte infiltration is seen. The white arrow shows a capillary vessel with an endothelium detached by attack of CD8 T-cells. CD8 IHC (immunohistochemistry), 400 × bar = 100 µm. (**C**) Macroscopic findings in the pelvis in autopsy on POD 196. A pale uterus with atrophy (yellow triangles) and left hydrosalpinx (white arrow) were observed. Ut, Uterus; Ov, Ovary; Od, Oviduct. (**D**) Intraoperative indocyanine green (ICG) fluorescence imaging of the transplanted uterus in autopsy. Enhancement of the uterus (yellow triangles) was absent in imaging. Ut, Uterus. (**E**) Histopathological findings of the grafted vessel in autopsy. Fibrotic occlusion with hemosiderin deposition is seen at the grafted vessel. Elastica van Gieson (EVG) stain, 100 × bar = 500 µm. (**F**) Histopathological findings of the donor’s left oviduct in autopsy. The donor’s grafted oviduct showed dilation and lymphocyte infiltration and vacuolar alteration (white arrow), which suggests mild rejection. Bar = 100 µm.

**Figure 3 jcm-08-01572-f003:**
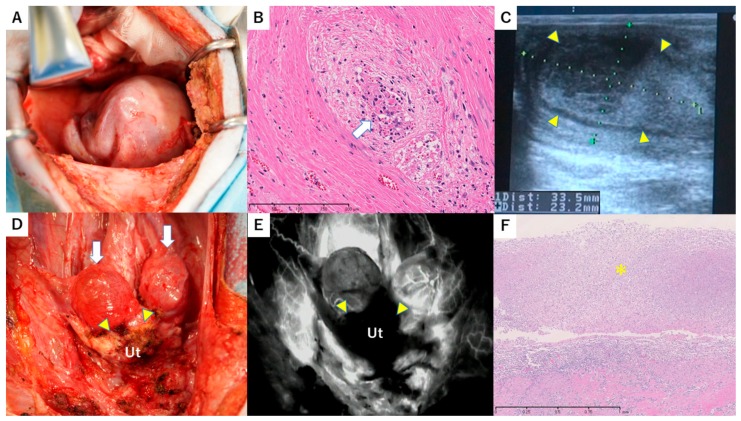
Macroscopic and histopathological findings in case 2. (**A**) Macroscopic findings in the pelvis in second-look surgery on POD 35. A markedly swollen uterus was observed. (**B**) Histopathological findings of the uterine myometrium in second-look surgery on POD 35. A small vessel in the myometrial interstitium showed severe endotheliitis. Due to lymphocytes infiltration, the endothelium is desquamated (white arrow). H&E stain, 200 × bar = 200 µm. (**C**) Transabdominal ultrasonography of the long axis of the uterine body on POD 97. An enlarged uterine body (33.5 × 23.2 mm: long axis × anteroposterior diameter) with a pooled abscess (yellow triangles) in the uterine cavity was found. (**D**) Macroscopic findings in the pelvis in autopsy on POD 126. A whitish atrophic uterus (yellow triangles) and bilateral hydrosalpinx (white arrow) were observed. Ut, uterus. (**E**) ICG fluorescence imaging of the transplanted uterus in autopsy. Enhancement of the grafted uterus (yellow triangles) was absent in imaging. Ut, uterus. (**F**) Histopathological findings of the removed uterus in autopsy on POD 196. The uterus had no endometrium. Neutrophilic exudate is seen, which suggests infection (*). On the bottom half, stromal tissue shows lymphocyte infiltration with capillary endotheliitis, indicating continuous rejection. Bar = 1mm.

**Figure 4 jcm-08-01572-f004:**
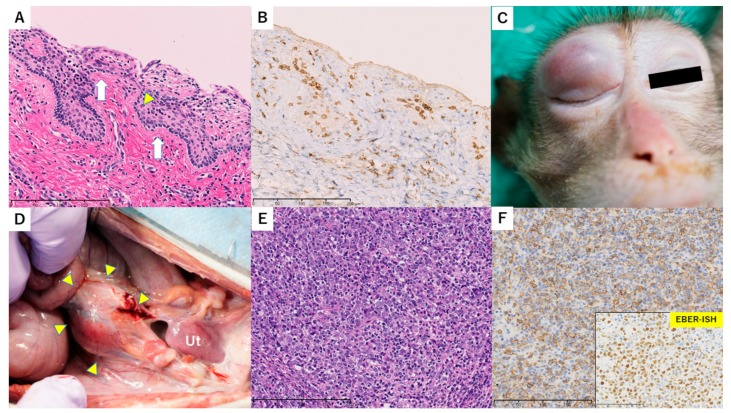
Macroscopic and histopathological findings in case 3. (**A**,**B**) Histopathological findings of a biopsy specimen from the uterine cervix on POD 83. Lymphocytes infiltrated the epithelium and stroma with mild vacuolar alteration (white arrow). A Civatte body was seen (yellow triangle). H&E stain, 200×. (**A**). Most of these lymphocytes were positive for CD8. CD8-IHC, 200×. (**B**) Bar = 200 µm. (**C**) The right upper eyelid of this animal was swollen. (**D**) Macroscopic findings in the pelvis in autopsy on POD 140. A reddish uterus of normal size was observed in the pelvis and a 4 cm tumor was present at the site of the abdominal aorta (yellow triangles). Ut, uterus. (**E**,**F**) Histopathological findings of a resected nodular tumor, showing monomorphic large-sized lymphoid cells with abundant nucleoli. H&E stain, 200x. (E). Bar = 100 µm. These atypical cells were positive for CD20 and EBER-ISH (window). (F). Bar = 200 µm.

**Figure 5 jcm-08-01572-f005:**
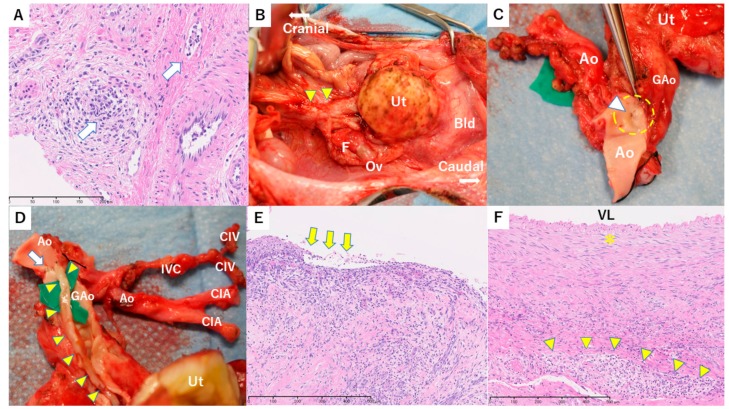
Macroscopic and histopathological findings in case 4. (**A**) Histopathological findings of a uterine tissue biopsy on POD 41. Perivascular lymphocyte infiltration and endotheliitis (white arrow) with perivascular fibrosis were clearly observed. Bar = 200 µm. (**B**) Macroscopic findings of the abdomen in autopsy on POD 104, showing a whitish swollen transplanted uterus. Severe adhesion at a site surrounding vascular anastomosis was present (yellow angle). The right adnexa was detected, but the left ovary and oviduct were not detected due to severe adhesion. Ov, ovary; Ut, uterus; F, fimbria; Bld, bladder. (**C**) The site of vascular anastomosis between aortas of recipient and donor. The aorta of the recipient was dissected and observation of the anastomotic site (yellow circle) showed that almost all of the lumen of the grafted aorta of the donor was stenosed, resulting in a pin hole (white triangle). Ao, aorta; GAo, grafted aorta; Ut, uterus. (**D**) Removed grafted (donor) and recipient vessels. The grafted aorta was along the long axis and intravascular thrombi were found from the anastomotic site (white arrow) to the grafted common iliac arteries (yellow triangles). Ao, aorta; Gao, grafted aorta; IVC, inferior vena cava; CIA, common iliac artery; CIV, common iliac vein; Ut, uterus. (**E**) Histopathological findings in the cervix of the removed uterus. Severe endotheliitis and epithelial desquamation (yellow arrows) were seen in the cervix. Bar = 500 µm. (**F**) Histopathological findings of the removed grafted aorta. Lymphocytes did not attack the endothelium (*) of the aorta directly; however, its feeding vessels in the vascular adventitia showed severe endotheliitis and obstruction (yellow triangles). Bar = 500 µm. VL, vascular lumen.

**Figure 6 jcm-08-01572-f006:**
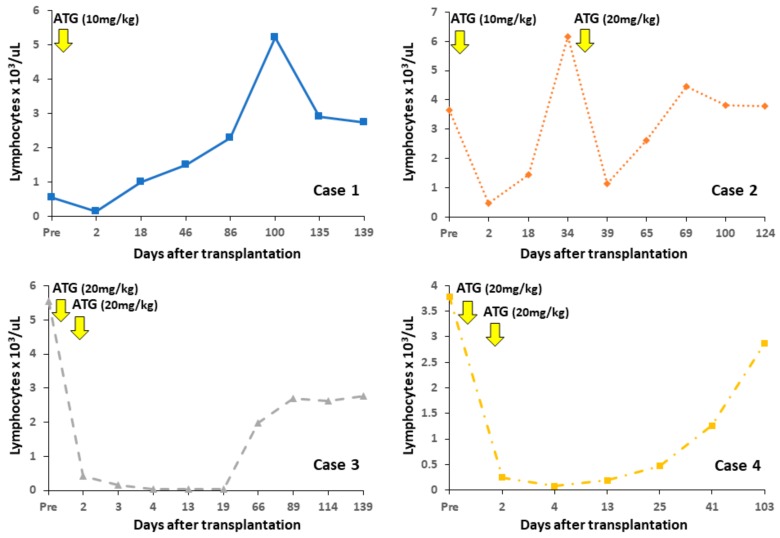
Changes in peripheral lymphocyte counts. Lymphocyte counts decreased after administration of antithymocyte globulin (ATG). The nadir lymphocyte counts in cases 3 and 4 were prolonged by differences in ATG doses, in comparison with counts in cases 1 and 2.

**Figure 7 jcm-08-01572-f007:**
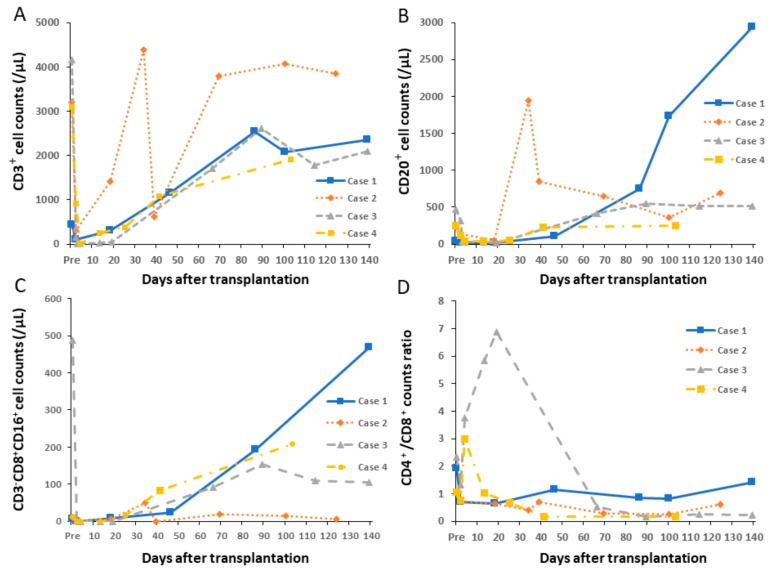
Changes in peripheral cell counts and ratio. (**A**) CD3^+^; (**B**) CD20^+^ and (**C**) NK cell^+^ counts decreased immediately after induction treatment, and then gradually increased. (**D**) The CD4^+^/CD8^+^ ratio showed a tendency to decrease.

**Figure 8 jcm-08-01572-f008:**
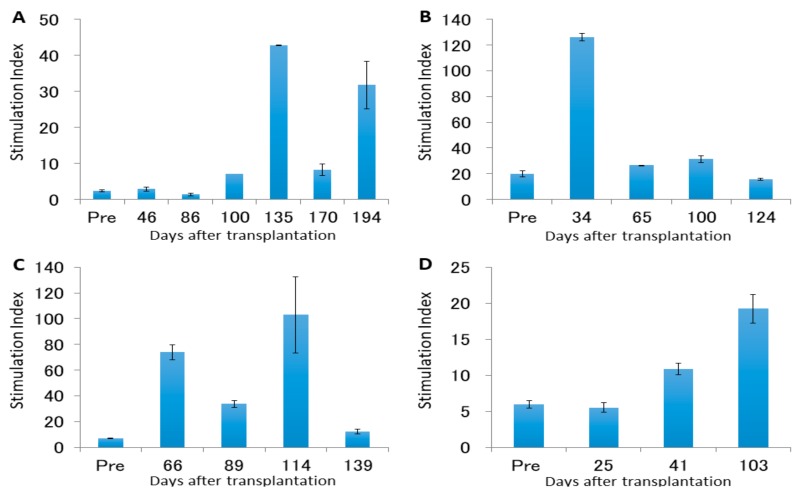
Mixed lymphocyte reaction (MLR) stimulation indexes pre- and post-transplantation. (**A**) Case 1. (**B**) Case 2. (**C**) Case 3. (**D**) Case 4.

**Figure 9 jcm-08-01572-f009:**
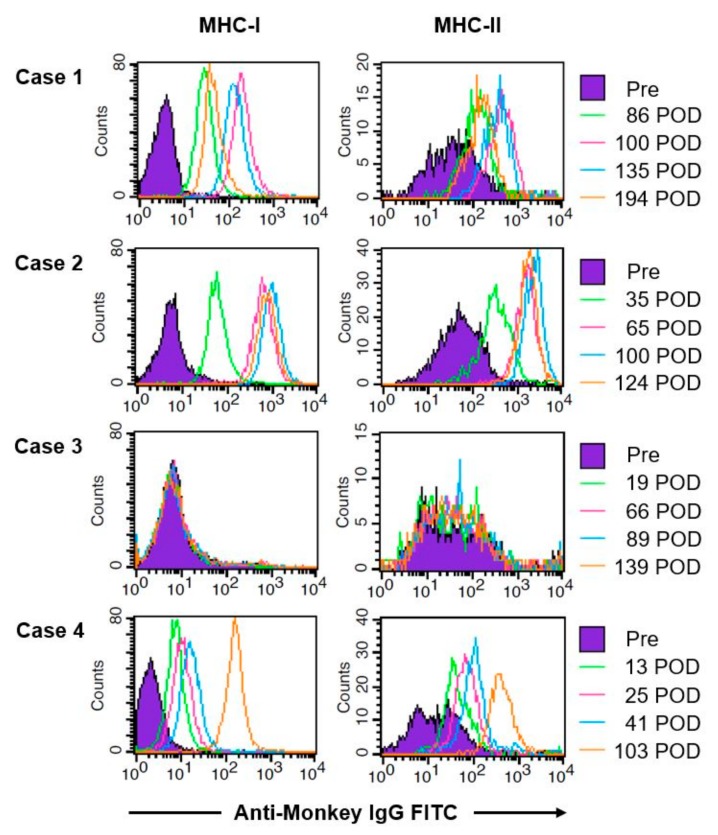
Changes of donor-specific antibodies (DSAs). Pre- and post-transplantation DSAs were produced against MHC class I and II in all animals except for case 3. The titers of DSAs increased thereafter.

**Table 1 jcm-08-01572-t001:** Summary of overall outcomes in recipients.

Case	Survival Days	Recovery of Menstruation	Rejection Episode	DSA Production	MLR Stimulation Indexes (Pre-Transplant; Maximum)	Cause of Sacrifice	Major Pathological Findings in Autopsy
1	196	Temporary	Severe	Positive	2.40; 40.0	Graft atrophy, chronic rejection	Atrophic uterus with hyalinized fibrosis and no endometrium Grafted vessels with fibrous occlusion and lymphocytes infiltration
DSA2	126	Temporary	Severe	Positive	19.9; 125.0	Graft atrophy, chronic rejection	Atrophic uterus with hyalinized fibrosis and uterine infection Uterine stroma with lymphocytes infiltration and capillary endotheliitis
3	140	None	Mild– Moderate	Negative	7.0; 33.7	Lower limbs paralysis	Normal-sized uterus with mild lymphocytes inflammation and a small number of Civatte bodies in the cervix Nodular tumors diagnosed as PTLD (diffuse large B-cell lymphoma)
4	104	None	Severe	Positive	5.9; 19.3	Swollen graft, chronic rejection	Whole uterus with severe endotheliitis, cervix with epithelial desquamation, and corpus with obstructive vasculitis and focal necrosis Feeding vessels in grafted aorta adventitia with severe endotheliitis and obstruction

DSA; Donor-specific antibody; MLR; Mixed lymphocyte reaction.

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
