# Peer review of "Long-Term Outcome and Rejection After Allogeneic Uterus Transplantation in Cynomolgus Macaques"

_jcm, 2019, doi:10.3390/jcm8101572_

Round 1

Reviewer 1 Report

I read with interest the work entitle « long-term outcome and rejection after allogeneic uterus transplantation in cynomolgus macaques ». This work deals with a preclinical animal models of uterus transplantation using allotransplantation in four cynomolgus macaques.  Indeed, all data were new and very useful for human application for uterus transplantation. Macaque animal models were very rare. I was impressed by the experimental model and design study. Congratulations.

I have only one major pitfalls. The results showed successively each case results such case reports. I think that authors should display results all together, with mean or median expression for continuous variables, such as a case series, not each case separately.

Author Response

Comments: I read with interest the work entitle « long-term outcome and rejection after allogeneic uterus transplantation in cynomolgus macaques ». This work deals with a preclinical animal models of uterus transplantation using allotransplantation in four cynomolgus macaques.  Indeed, all data were new and very useful for human application for uterus transplantation. Macaque animal models were very rare. I was impressed by the experimental model and design study. Congratulations.

Response: We are grateful for the positive comments on our work.

Comments: I have only one major pitfalls. The results showed successively each case results such case reports. I think that authors should display results all together, with mean or median expression for continuous variables, such as a case series, not each case separately.

Response: We are grateful to for the useful suggestions that results of all cases should be displayed together. We of course considered this when we wrote this manuscript. However, we could not appropriately summarize these cases with median expression for continuous variables because these cases had different postoperative courses. We decided that it would be better for the readers to understand the outcomes clearly with description of each postoperative course. We would be pleasured if the reviewer could understand our situation.

Reviewer 2 Report

The authors have done extensive work but unfortunately no success. The study may be helpful for future research on uterus transplantation.

Wondering in humans, who will be the donor for allogeneic UTx, there may be recipients. Whether this kind of study has real impact on human race?

Whether UTx is possible between different species; maybe between macaques and baboons?

Author Response

Comments: The authors have done extensive work but unfortunately no success. The study may be helpful for future research on uterus transplantation.

Response: We are grateful for the positive comments on our work.

Comments: Wondering in humans, who will be the donor for allogeneic UTx, there may be recipients. Whether this kind of study has real impact on human race?

Response: We are sorry that we could not understand the intentions of the reviewer. In human, the potential donors of UTx are the relatives and third parties. However, we believe we should mention it in this manuscript because this has no relation to this basic study. We also believe that these data will be useful and have impact for human study because non-human primates are anatomically and physiologically similar to humans.

Comments: Whether UTx is possible between different species; maybe between macaques and baboons?

Response: This may be possible, but we don’t assume xenogeneic transplantation even if they are between non-human primates. We believe we should mention it in this manuscript because this has no relation to this study.